# Competing Endogenous RNA of Snail and Zeb1 UTR in Therapeutic Resistance of Colorectal Cancer

**DOI:** 10.3390/ijms22179589

**Published:** 2021-09-03

**Authors:** Nam Hee Kim, Sang Hyun Song, Yun Hee Choi, Kyu Ho Hwang, Jun Seop Yun, Hyeeun Song, So Young Cha, Sue Bean Cho, Inhan Lee, Hyun Sil Kim, Jong In Yook

**Affiliations:** 1Department of Oral Pathology, Yonsei University College of Dentistry, Seoul 03722, Korea; MIGO77@yuhs.ac (N.H.K.); SSH407@yuhs.ac (S.H.S.); cyunee@nate.com (Y.H.C.); tinogo@yuhs.ac (K.H.H.); YJS8714@yuhs.ac (J.S.Y.); 0212HEA@yuhs.ac (H.S.); sysh901@yuhs.ac (S.Y.C.); chosuebean@gmail.com (S.B.C.); 2Research Division, miRCore, Ann Arbor, MI 48105, USA; inhanlee99@gmail.com

**Keywords:** ceRNA, epithelial mesenchymal transition, Snail, ZEB1, colorectal cancer, therapeutic resistance

## Abstract

The epithelial-mesenchymal transition (EMT) comprises an important biological mechanism not only for cancer progression but also in the therapeutic resistance of cancer cells. While the importance of the protein abundance of EMT-inducers, such as Snail (SNAI1) and Zeb1 (ZEB1), during EMT progression is clear, the reciprocal interactions between the untranslated regions (UTRs) of EMT-inducers via a competing endogenous RNA (ceRNA) network have received little attention. In this study, we found a synchronized transcript abundance of Snail and Zeb1 mediated by a non-coding RNA network in colorectal cancer (CRC). Importantly, the *trans*-regulatory ceRNA network in the UTRs of EMT inducers is mediated by competition between tumor suppressive miRNA-34 (miR-34) and miRNA-200 (miR-200). Furthermore, the ceRNA network consisting of the UTRs of EMT inducers and tumor suppressive miRs is functional in the EMT phenotype and therapeutic resistance of colon cancer. In The Cancer Genome Atlas (TCGA) samples, we also found genome-wide ceRNA gene sets regulated by miR-34a and miR-200 in colorectal cancer. These results indicate that the ceRNA networks regulated by the reciprocal interaction between EMT gene UTRs and tumor suppressive miRs are functional in CRC progression and therapeutic resistance.

## 1. Introduction

The epithelial-mesenchymal transition (EMT) comprises an important biological program involving development and cancer progression [1]. While earlier EMT studies had focused on phenotypic conversion from an epithelial to a mesenchymal state, recent evidence indicates that EMT plays a critical role in cancer stemness, metabolic reprogramming, immune evasion, inflammation and therapeutic resistance of cancer cells [2]. Snail (SNAI1) and ZEB1 are well-known inducers of cancer EMT, and their protein abundance is transcriptionally or post-translationally regulated by many oncogenic signaling pathways, such as canonical Wnt and TGF-β [3,4,5,6].

MicroRNAs (miRs) are ubiquitous post-transcriptional regulators that impact RNA stability and the rate of translation by pairing to complementary sites within target RNAs [7,8,9,10]. The UTRs of Snail and Zeb1 are targeted by tumor suppressive miR-34a and miR-200, respectively [11]. Typically, an miR binding to an untranslated region (UTR) represses the protein translation of mRNA. In turn, the UTRs of mRNA transcripts have an miRNA-binding motif suppress protein expression in *cis*-regulation. However, the competing endogenous RNA (ceRNA) hypothesis posits an inverse role for the UTRs involving the *trans*-regulatory crosstalk of UTRs through common miRs shared and sequestered by competing RNAs [12,13]. According to the ceRNA hypothesis, non-coding RNAs (such as UTRs, long non-coding RNA (lncRNA) and pseudogenes) regulate other RNA transcripts by competing for shared miRs. For example, PTEN and VCAN share miR-136 and miR-144 binding sites in their UTRs and, therefore, compete for these miRNAs, resulting in an RNA interaction between PTEN and VCAN [12,13]. Therefore, the ceRNA hypothesis provides a novel layer of RNA transcript interaction via miRNA competition independent of its protein coding function. We have previously reported that the overexpression of Wnt gene UTRs, including Wnt1, LRP6, CTNNB1 (beta-catenin) and Snail UTRs, with miR-34a binding sites increased the endogenous Snail expression [3,14,15], suggesting the existence of a *cis*- and *trans*-regulatory network between UTRs sharing miR-34a binding sites [16].

Although the ceRNA hypothesis provides an interesting role for an miR and non-coding RNA, including UTRs, pseudogenes, antisense transcripts and long non-coding RNA, ceRNA regulatory networks in EMT genes are not well-known. In this study, we found a synchronous abundance of Snail and Zeb1 transcripts in human cancer samples and an inverse abundance between tumor suppressive miR-34a and miR-200 to EMT inducers. Although Snail’s UTR does not have an miR-200 binding site, we found an indirect ceRNA interaction between the UTRs of EMT inducers and tumor suppressive miRs. Further, the UTRs of EMT inducers or tumor suppressive miRs play a role in the therapeutic resistance of colorectal cancer cells. Our results suggest not only that a novel ceRNA network between Snail and Zeb1 is regulated by miR-34a and miR-200, but, conversely, that the non-coding EMT gene UTRs dynamically regulate the tumor suppressive miRs.

## 2. Results

### 2.1. Coincident Snail and Zeb1 Transcript Abundance in Human Cancer

While Snail and Zeb1 constitute key EMT inducers, their relative transcript abundance in human cancer samples has not been well-studied. Examining Snail and Zeb1 transcripts in The Cancer Genome Atlas (TCGA) samples, we found their abundance to be significantly correlated in COADREAD (colorectal cancer), and that patients with a high abundance of Snail and Zeb1 show poor survival prognosis (Figure 1a), as do patients with a high abundance of either Snail or Zeb1 (Appendix A). We also found a concordant abundance of Snail and Zeb1 transcripts in BRCA (breast cancer), LUAD (lung adenocarcinoma), LUSC (lung squamous cell carcinoma) and HNSC (head and neck squamous cell carcinoma) samples (Appendix A), suggesting crosstalk between the Snail and Zeb1 transcripts in human cancer.

Exploring the hypothesis that Snail and Zeb1 show accordant transcript abundance, we chose HCT116, a colorectal cancer cell line with high expression levels of Snail and Zeb1. A knockdown of Snail or Zeb1 downregulates the endogenous protein and mRNA transcript levels of Zeb1 and Snail, in tandem (Figure 1b,c). To determine whether the reciprocal regulation between Snail and Zeb1 is independent of the protein level, we next generated protein-coding null cells for Snail and Zeb1 with CRISPR-Cas9 in HCT116 cells (Appendix A). As expected, Snail and Zeb1 null cells showed epithelial phenotypes by means of increased E-cadherin expression, decreased migratory potential and increased susceptibility to paclitaxel compared to parental HCT116 cells (Appendix A). To examine the crosstalk between the Snail and Zeb1 transcript independent of the protein-coding function, we knocked down the Snail or Zeb1 transcript with siRNA in Snail or Zeb1 null cells and determined their transcript levels (Figure 1d). Indeed, the knockdown of the Snail or Zeb1 transcript suppressed the other in those null cells, suggesting crosstalk between the Snail and Zeb1 transcripts via non-coding interactions, such as UTR and miR.

### 2.2. Snail and Zeb1 Are Regulated by Shared miRs

Previously, we and others reported that 3′ UTRs of Snail and Zeb1 contain tumor suppressive miR binding sites miR-34a and miR-200, respectively (Figure 2a) [3,17,18,19,20]. To examine whether these tumor suppressive miRs are involved in the UTR interaction between Snail and Zeb1, we next overexpressed miR-34a or miR-200 together with luciferase vectors having a Snail or Zeb1 UTR, then measured the reporter activity and protein abundance in HCT116 cells. As expected, miR-34a or miR-200 suppressed Snail and Zeb1 UTR activity along with their endogenous protein abundances (Figure 2b,c). Notably, miR-200 also suppressed Snail’s UTR activity and protein abundance, although there is no putative binding site for miR-200, indicating that those UTRs competitively bind to tumor suppressive miRs. To directly test this notion, we overexpressed the miR-34a or miR-200 sponge together with the Snail or Zeb1 UTR luciferase constructs and measured the reporter activity. Interestingly, the miR-34a or miR-200 sponge increased the Snail and Zeb1 UTR activities as well as the endogenous protein abundance (Figure 2d,e). The overexpression of the miR-34a or miR-200 sponge increased the migratory potential and therapeutic resistance against the paclitaxel of HCT116 cells (Figure 2f,g).

Given the concordant transcript abundance of Snail and Zeb1, we next examined the miR-34a and miR-200 abundance in colon cancer samples. In TCGA samples, we found the miR-34a and miR-200 abundance to be highly correlated in the COADREAD samples (Figure 2h). This correlation was also found in BRCA, LUAD, LUSC and HNSC (Appendix A). These results suggest that miR-34a and miR-200 regulate each other similarly to the UTRs of Snail and Zeb1. To explore this possibility, we overexpressed the tumor suppressive miR expression vectors or sponge vectors and determined the transcript abundances. Indeed, the miR-34a and miR-200 abundances were tightly correlated (Figure 2i,j).

### 2.3. UTRs of Snail and Zeb1 Can Be Regulatory Factors in a ceRNA Network

As the UTR transcript of EMT genes and tumor suppressive miRs were tightly co-regulated, we hypothesized that the UTRs of Snail and Zeb1 compete with tumor suppressive miRs and are reciprocally regulated by a *trans*-regulatory network (Figure 3a). Supporting this notion, the overexpression of a Snail or Zeb1 UTR decreased both the miR-34a and miR-200 abundance while increasing the migratory potential of HCT116 cells (Figure 3b,c). To determine whether the miRs are critically involved in the *trans*-regulatory network, we next overexpressed the EMT UTR constructs in parental or Dicer-null HCT116 cells, then measured the transcript and endogenous protein abundance of Snail and Zeb1. The *trans*-regulation of the Snail and Zeb1 UTR transcript disappeared and endogenous protein abundances were unchanged in Dicer-null HCT116 cells (Figure 3d,e). To examine the possibility of an endogenous protein of Snail and Zeb1, we next constructed mCherry expression vectors with UTRs of Snail or Zeb1 (Figure 3f). Co-transfecting mCherry in combination with luciferase expression vectors in Snail-null or Zeb1-null cells, we observed *trans*-regulation between the UTRs of Snail and Zeb1 independent of a protein-coding function (Figure 3g). The overexpression of Snail or Zeb1 UTRs was sufficient to increase the migratory potential of HCT116 cells (Figure 3h). These results support a *trans*-regulatory network between Snail and Zeb1 independent of the protein-coding function via tumor suppressive miRs, at least in part.

### 2.4. UTRs of Snail and Zeb1 Induce Cancer Stem Cell-Like Properties

It is well known that EMT induces the ‘stemness’ properties of cancer cells, including resistance to apoptosis, self-renewal, metabolic rewiring and survival of cancer stem cells [2,3,17,21,22,23,24,25]. To validate that the non-coding UTR of an EMT gene is functional, we generated inducible cells of a Snail or Zeb1 UTR in HCT116 cells to examine whether UTR expression affects anchorage-independent growth in a soft-agar assay. Interestingly, the inducible expression UTR of Snail or Zeb1 increased the number of spherical colonies compared to the negative control (Figure 4a). Furthermore, the overexpression of the UTR of EMT genes increased the expression of CD44, a marker of colorectal cancer stem cells (Figure 4b) [22,23,24], as well as therapeutic resistance against paclitaxel (Figure 4c) [26]. In protein coding null cells of Snail or Zeb1, the overexpression of UTR was sufficient to increase therapeutic resistance (Figure 4d). These results indicate that the UTRs of EMT genes are functionally independent of the protein coding function of Snail or Zeb1. To examine the reciprocal interaction between the Snail and Zeb1 UTR in vivo, we generated inducible cells of the Snail or Zeb1 UTR in 293 cells, then designed an experiment to examine the UTR interaction (Figure 4e). Indeed, the in vivo induction of the Snail or Zeb1 UTR increased the protein abundance as well as the transcript abundance in *cis* (Snail UTR vs. Snail transcript or protein; Zeb1 UTR vs. Zeb1 transcript or protein) and in *trans* (Snail UTR vs. Zeb1 transcript or protein; Zeb1 UTR vs. Snail transcript or protein) (Figure 4f,g).

### 2.5. Genome-Wide ceRNA Network of EMT Genes

Given our observations of a *trans*-regulatory network between Snail and Zeb1, we next expanded our ceRNA network search to EMT genes genome-wide. We chose 100 predicted co-target genes of miR-34a and miR-200 using TargetScan (http://www.targetscan.org (accessed on 9 February 2018)) according to the score sum of both miRs (Appendix A) and analyzed the correlation of those co-target genes in COADREAD samples. Interestingly, we found three gene sets clustered in clinical samples (Figure 5a). Furthermore, the Snail and Zeb1 transcripts were segregated, together with 21 other transcripts (Appendix A). Analyzing the transcript abundance of those genes and tumor suppressive miRs, we found transcript abundance to be inversely correlated with that of miR-34a and miR-200 (Figure 5b). We next screened the mRNA transcript abundance of those genes, choosing several genes expressed in HCT116 cells. We then knocked down Snail or Zeb1 using siRNA and examined the transcript abundance of those genes. Indeed, transcript abundance was suppressed by either a Snail or Zeb1 knockdown (Figure 5c). These results suggest the existence of a genome-wide *trans*-regulatory ceRNA network of EMT genes independent of the coding function of transcripts.

## 3. Discussion

EMT in cancer is a well-known biological program for cancer progression, accompanied by a loss of cell polarity, a reduced expression of basement membrane components, an enhanced catabolic metabolism and an increased propensity for metastasis [25,27,28,29,30]. While the EMT promotes phenotypic transition from epithelial to mesenchymal, transiently or stably, recent findings indicate that it also plays a critical role in cancer stemness, the inflammatory response and the acquisition of drug resistance [21]. Many EMT-promoting transcription factors, such as Snail (SNAI1), Slug (SNAI2), TWIST (TWIST1) and Zeb1 (ZEB1), can suppress epithelial genes directly or indirectly [3,17,31,32,33,34]. The expression of these genes is regulated by a transcriptional and post-translational modification within oncogenic signaling pathways [4,5,17,31,32]. For example, the canonical Wnt pathway stabilizes the Snail protein by the inhibition of ubiquitination and TGF-β promotes the transcription of EMT genes [3,28,35]. Recent studies have demonstrated that tumor suppressive miR-34a and miR-200 repress Snail and Zeb1 expression via direct targeting of their respective UTRs [3,17,18,19,20,36,37]. Although most EMT studies have been performed based on the protein-coding function of those genes, the role of the UTR of EMT-inducers is not well-understood. As the ceRNA hypothesis introduces an additional interesting layer within the biological network [10,12,13,38], we explored the role of a *trans*-regulating ceRNA network related to EMT genes and tumor suppressive miRs.

In this study, we found an miR-mediated *trans*-regulation ceRNA network involving Snail and Zeb1 UTRs resulting in a concordant abundance of EMT genes. Our observations have several implications for cancer progression and the therapeutic resistance of cancer cells. First, the UTR of an mRNA transcript is largely regarded as a negative *cis*-regulator of protein expression. We found that the UTRs of EMT genes compete with tumor suppressive miRs and function as a positive *trans*-regulator of transcript and protein abundance. For example, the overexpression of a Snail or Zeb1 UTR without a coding gene increased endogenous Snail and Zeb1 protein expression by competing with tumor suppressive miRs. Second, the interaction between an miR and a UTR is considered a weaker interaction biologically. However, we found that EMT transcripts were inversely correlated with tumor suppressive miRs in many sets of clinical cancer samples, indicating that ceRNA networks are more influential than previously assumed. Third, a ceRNA network of non-coding UTRs and miRs plays a role in the therapeutic resistance and EMT phenotype of cancer cells independent of the protein coding function. In turn, the non-coding RNA function is not confined to the negative regulation of gene expression. Lastly, we found a genome-wide ceRNA network of the co-target genes of tumor suppressive miRs. Note that only a few negative correlations were found among the co-target genes of tumor suppressive miRs (Figure 5b). Although we only chose 100 genes according to the sum score (Appendix A), the *trans*-regulatory network consists of several layers of reciprocal interactions, broadening the biological function of non-coding RNA. For example, we found another gene set including the tumor suppressor PTEN, suggesting that EMT genes are indirectly connected with PTEN via a ceRNA network.

Transcriptional landscape studies of functional DNA elements and comparative studies have revealed that a majority of transcripts consist of non-coding RNA harboring diverse and significant gene regulatory functions, especially in human diseases and cancer [39,40]. While we only exemplified the non-coding UTRs of EMT genes and tumor suppressive miRs, the *trans*-regulating ceRNA network is also regulated by other non-coding RNAs, such as long non-coding RNAs (lncRNAs), circular RNAs, pseudogenes and anti-sense transcripts [13].

In conclusion, we demonstrate that Snail and Zeb1 expression, in terms of transcript abundance as well as protein expression, is coordinated via tumor suppressive miRs in human cancer. As Snail and Zeb1 are well-known independent EMT inducers, our observations provide a *trans*-regulatory non-coding RNA network between EMT genes beyond the protein coding function. Further, the ceRNA network is functional in therapeutic resistance and tumor progression. Although our study only focused on EMT genes and tumor suppressive miRs, further study is needed to broaden our understanding of oncogenic or tumor suppressive ceRNA networks in human cancer.

## 4. Materials and Methods

### 4.1. Cells and Constructs

HCT116 wt (parental) and Dicer-null cells were kindly provided by Dr Vogelstein’s lab as described previously [3,31]. They were cultured in Dulbecco’s modified Eagle’s medium (DMEM) (Lonza, Basel, Switzerland) supplemented with 10% fetal bovine serum (FBS) (Life Technologies, Carlsbad, CA, USA) [4,41]. miR and UTR expression vectors were described previously [11,14]. To create the miR-34a and miR-200 sponge vector with three perfectly complementary miRNA binding sites, two complementary oligos (an upper and lower strand) were synthesized, annealed and inserted between the BamHI and NotI sites or Not1 and Xba1 sites into pcDNA3.1-lucifrease vector and pcDNA3.1-mCherry vector. The miR-34 and 200 sponge sequence were miR-34a sponge oligo 5′-GATCCACAACCAGCTAAGACACTGCCAAACAATCAGCTAATGACACTGCCTAAAGCAATCAGCTAACTACACTGCCTGC and miR-200 sponge oligo 5′-GGCCGCACAACCAGCTAAGACACTGCCAAACAATCAGCTAATGACACTGCCTAAAGCAATCAGCTAACTACACTGCCTT. Tetracycline-inducible Snail and Zeb1 UTR expression vectors were generated with the pTRIPZ lentiviral system (Open Biosystems, Huntsville, AL, USA) by replacing RFP. A pTRIPZ-Snail and Zeb1 UTR expression vector was used to generate retroviral stocks in 293 cells for infecting 293A and HCT116 cells. Stable UTR transfectants were obtained after selection with 1.0 µg/mL puromycin (Sigma, St. Louis, MO, USA). To obtain knockdown of Snail or Zeb1 expression in HCT116 cell, corresponding CRISPR/Cas9-assisted plasmids were used. Expression vectors for guide RNA (pRGEN-sgRNA-U6) and Cas9 gene (pRGEN-Cas9-CMV) were purchased from ToolGen (ToolGen inc., Seoul, Korea). Cells were seeded into a 96-well plate to isolate cell colonies at 48 h after CRISPR plasmid treatment. Knockout cells were screened using Western blot and genomic DNA sequencing.

### 4.2. Snail and Zeb1 UTR Reporter Assays

The predicted miR-34a and miR-200 target sites in the 3′UTR of Snail and Zeb1 were identified as described previously [11,14]. A luciferase expression construct with multiple cloning sites for UTRs was used as described previously [3,14]. The 3′UTRs of Snail and Zeb1 were amplified from the genomic DNA of MCF-7 cells and sub-cloned into the BamH1 and NotI sites downstream of luciferase. For targeting of endogenous miR-34a and miR-200, cells were transiently transfected with 3′ UTR reporter constructs (1–5 ng) and miR vectors that induce expression or suppression (500 ng) using Lipofectamine 2000 (Invitrogen, Waltham, MA, USA). The activity of 3′ UTR reporter constructs was normalized to the activity of the cotransfected SV40-renilla luciferase construct (2 ng) (Promega, Madison, WI, USA). Cells were lysed 48 h after transfection and the relative ratio of renilla luciferase to firefly luciferase activity was measured in a dual luciferase assay (Promega, Madison, WI, USA).

### 4.3. Quantitative-PCR (qPCR)

HCT116 cells were transiently transduced with siRNA for Snail and Zeb1. SiRNA duplexes directed against human Snail and Zeb1 were obtained from Santa Cruz Biotechnology. The relative expression levels of Snail, Zeb1, CORO1C, RGS17, ELMOD1, IDS and QKI transcript were determined using real time quantitative PCR as described previously [14]. Briefly, total RNA of the cells was isolated with Trizol (Invitrogen, Waltham, MA, USA), and cDNA was synthesized using random hexamer reverse transcription primer (Intron, Seoul, Korea). Transcript levels were detected using the 7300 Real-Time PCR System and SYBR Green (Applied Biosystems, Foster, CA, USA) with qPCR primer (Table 1) and both then normalized to the levels of GAPDH; the relative log2 expression levels in cancer tissue compared with paired normal tissue were then calculated. For quantitative analysis of mature miR-34a and miR-200 levels, human TaqMan miRNA assay kits (Applied Biosystems, assay ID 000426 for miR-34a, ID 000502 for miR-200 and ID001973 for U6 snRNA, Foster, CA, USA) were used for reverse transcription with specific primers and qPCR was performed with corresponding probes (*n* = 3). The expression of mature miR-34a and miR-200 ΔCt values from each sample were calculated by normalizing with U6 expression values.

### 4.4. Western Blot Analysis

The protein levels of Snail and Zeb1 were detected using Western blot analysis of whole cell lysate with Triton X-100 with antibodies direct E-cadherin (610181, Invitrogen, Waltham, MA, USA), Snail (3895s, Cell Signaling Technology, Danvers, MA, USA), Zeb1 (3396, Cell Signaling Technology, Danvers, MA, USA) and Tubulin (Labfrontier Co., Ltd., Seoul, Korea).

### 4.5. Transwell Assay

For migration assays, HCT116 or Snail or Zeb1 null HCT cells by CRISPR-Cas9 were transfected with 3′ UTR vectors and miR vectors that induce overexpression or suppression of miRs with Lipofectamine. After 24 h, cells were trypsinized, 10^5^ cells were seeded into Transwell inserts (5.0 μm pore) (BD Biosciences, Palo Alto, CA, USA), and the number of cells migrating to the basal side of the membrane insert was determined at 72 h. Migrated cells were calculated as area by Image J 1.52 software (National Institutes of Health, Bethesda, MD, USA) through the particle analysis method.

### 4.6. Colony Formation Assay and Soft Agar Assay

For drug resistance assay, colony formation assays were performed with 3′ UTR vector, miR expression or suppression vector transfected cells. The 5 × 10^4^ cells were seeded in 6-well plates. After 3 days of exposure to the 5 μM paclitaxel, the cells were washed with PBS and cultured in normal culture medium for an additional 10–14 days to determine clonogenic survival. After crystal violet (0.5% *w*/*v*) staining, colonies of more than 50 cells were counted under stereomicroscope. The number of colonies in 5 randomly chosen fields was determined under high power stereomicroscope. Soft agar colony formation assay was performed as described previously [42]. Briefly, Tet-inducible UTR expression HCT116 was suspended at 1 × 10^4^ cells per 6-well plate with 1 mL of 0.3% low-melting agar in 2X DMEM containing 20% FBS and overlaid on a layer of 1 mL of 1% agar in the same medium. After 2 weeks of incubation, viable colonies that contained >50 cells were counted from five fields with a stereomicroscope. Representative colonies were photographed, and two independent experiments were performed.

### 4.7. Flow Cytometry

Cells were dissociated with trypsin and resuspended in PBS with 3% BSA. Cells were labeled with FITC (fluorescein isothiocyanate) conjugated monoclonal anti-CD44 antibody (BD Biosciences, Palo Alto, CA, USA) at 4 °C for 30 min. Analysis was performed using a BD FACS Aria II.

### 4.8. In Vivo Xenograft Assay

Female athymic nude mice (6-week-old) were used for xenograft assays into flank subcutaneous tissue. A total of 293 cells expressing Tet-inducible UTR (1 × 10^6^) were resuspended in 100 μL of PBS and injected into flank subcutaneous tissue. When the tumors reached an average of 1000 mm^3^, mice were given intraperitoneal administration of 5 mpk doxycycline three times. The mice were sacrificed, and the tumors were extracted for paraffin-embedding. Tissue lysate was isolated using Trizol from tumor cryosections. Transcript levels of Snail and Zeb1 in tumor samples were detected using qPCR.

### 4.9. Gene Expression Analysis of Clinical Samples

Publicly available mRNASeq and miRNASeq data including long-term survival information from TCGA was downloaded (https://gdac.broadinstitute.org (accessed on 2 February 2017)). The dataset (data version 2016_01_28) included colorectal adenocarcinoma (COADREAD, mRNA 623 samples, miRNA 295 samples, mutation 223 samples), breast cancer (BRCA, mRNA 1093 samples, miRNA 755 samples, mutation 981 samples), lung adenocarcinoma (LUAD, mRNA 515 samples, miRNA 450 samples, mutation 230 samples), lung squamous cell carcinoma (LUSC, mRNA 501 samples, miRNA 342 samples, mutation 178 samples) and head and neck squamous cell carcinoma (HNSC, mRNA 520 samples, miRNA 486 samples, mutation 279 samples). The illuminahiseq_rnaseqv2-RSEM_genes_normalized (MD5) was log2 transformed. Mutational information and clinical data for survival analysis were obtained using Mutation_Packager_Calls (MD5) in the level 3 dataset and Clinical_Pick_Tier1 (MD5) in the level 4 dataset. For unsupervised hierarchical cluster analysis, Ward linkage method was used together with the Pearson distance for both sample and gene clustering. For survival analysis, low and high expression groups were determined from the lowest *p* value between 10–90 percentile. The high and low abundance subsets were determined based on the median transcript abundance of Snail and Zeb1, yielding groups with the most significant differences in survival based on the log-rank test described previously [42,43]. The R package was used to generate a Kaplan–Meier plot (survival), correlation plot (corrplot) and heatmap plot (gplots).

### 4.10. Statistical Analysis

Statistical significance of reporter assays and qPCR were determined using an independent sample *t*-test, and Mann–Whitney test was performed to determine significant differences in colony formation and Transwell assay among group means for non-parametric statistical hypothesis using IBM SPSS Statistics (Released 2017 for windows, Version 25.0 Armonk, NY: IBM Corp, USA). Differences were considered significant when the *p* value was less than 0.05 or 0.01 as indicated in the text.

## Figures and Tables

**Figure 1 ijms-22-09589-f001:**
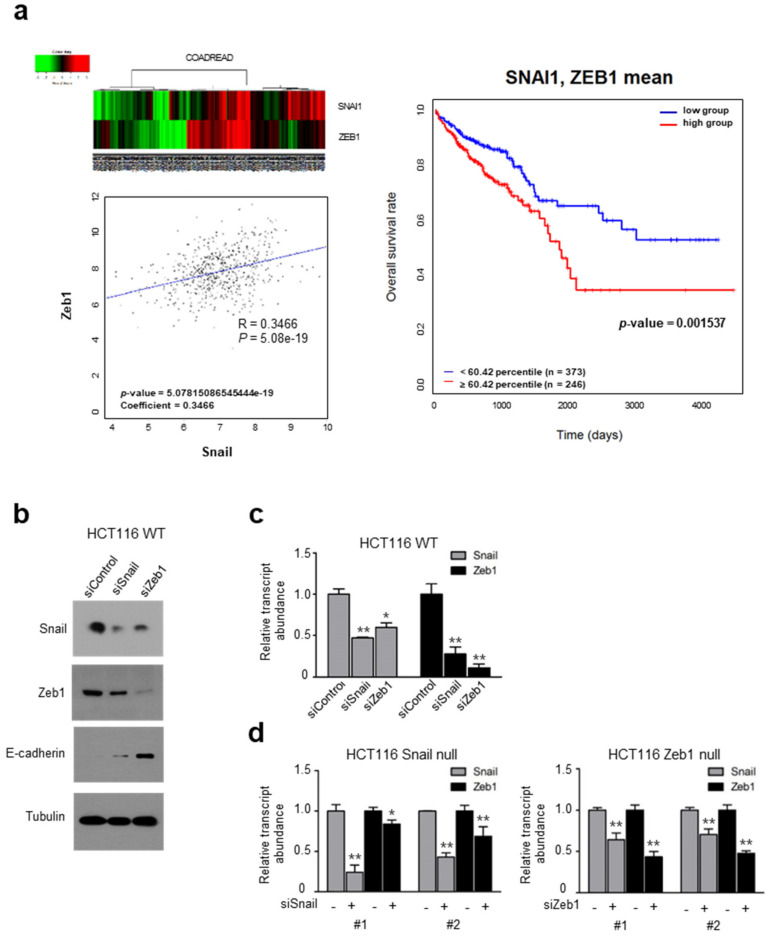
Concordant transcript abundance of Snail and Zeb1: (**a**) A supervised hierarchical clustering based on Snail and Zeb1 expression profiles (left upper panel) and Pearson correlation scatter plots of Snail and Zeb1 transcript in colorectal cancers (left lower panel). Kaplan–Meier survival graph for colorectal cancer patients with lower (blue line) or higher (red line) expression levels of Snail and Zeb1 on the basis of their transcript abundance at an optimal threshold, indicated by percentile number (right panel). In the heatmap, red denotes higher relative expression, whereas green indicates lower relative expression, with degree of color saturation reflecting the magnitude of the log expression signal; (**b**) Immunoblot analysis of Snail, Zeb1 and E-cadherin in HCT116 after siRNA-mediated knockdown of Snail and Zeb1. Tubulin was used as loading control. Representative blots are shown from at least two independent experiments; (**c**) Quantitative PCR (qPCR) analysis of Snail and Zeb1 after siRNA-mediated knockdown of Snail and Zeb1 in HCT116 (* *p* < 0.05, ** *p* < 0.01 compared with siControl, *t*-test); (**d**) In HCT116 of which Snail and Zeb1 were knocked out, respectively, by CRISPR-Cas9 systems, qPCR analysis of Snail and Zeb1 was performed after siRNA-mediated knockdown of Snail and Zeb1 (* *p* < 0.05, ** *p* < 0.01 compared with siControl, *t*-test).

**Figure 2 ijms-22-09589-f002:**
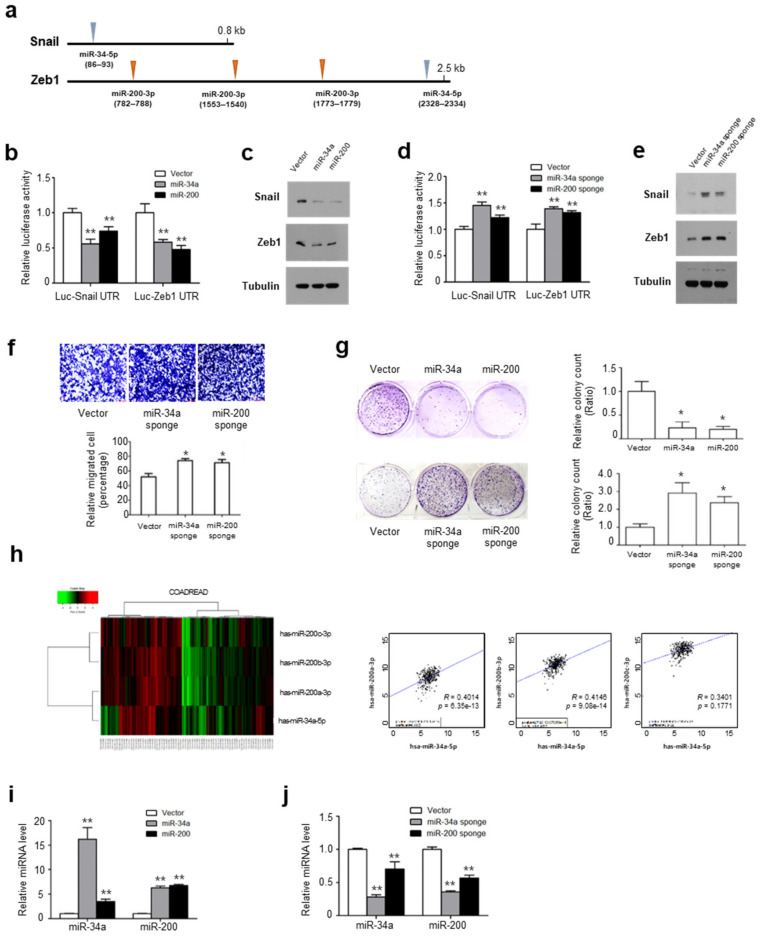
Snail and Zeb1 shared regulatory miR-34a and miR-200: (**a**) Potential matching target sites of miR-34a and miR-200 in the UTR regions of Snail and Zeb1; (**b**) Snail and Zeb1 UTR reporter activity in the HCT116 cells that were transient transfected pMSCV-miR-34a or miR-200 expression vectors. Reporter activity was normalized to the activity of SV40 Renilla (** *p* < 0.01 compared with control, *t*-test); (**c**) Immunoblot analysis of endogenous Snail and Zeb1 level after transfection of miR-34a or miR-200 vectors. Representative blots are shown from at least two independent experiments; (**d**) Snail and Zeb1 UTR reporter activity in HCT116 cells transfected with miR-34a or miR-200 sponge vector (** *p* < 0.01 compared with control, *t*-test); (**e**) Immunoblot analysis of endogenous Snail and Zeb1 level after inhibition of miR-34a or miR-200 by the transfection of sponge vector. Representative blots are shown from at least two independent experiments; (**f**) Representative migration images in HCT116 cells transfected with miR sponge vector (upper panel). Migratory activities after function loss of miR-34a and miR-200. Results are representative of five independent experiments (lower) (* *p* < 0.05 compared with vector, Mann–Whitney test); (**g**) Clonogenic survival assay against paclitaxel in HCT116 cells with transfection of miRs overexpression vector or miRs sponge vector (* *p* < 0.05 compared with control, Mann–Whitney test); (**h**) Unsupervised hierarchical clustering based on miR-34a and miR-200 expression profiles (right panel) and Pearson correlation scatter plots (left panel) of miR-34a and miR-200 transcript in colorectal cancers; (**i**,**j**) Quantitative PCR analysis of miR-34a or miR-200; (**i**) After transfection with an miR-34a and miR-200 expression vector and (**j**) After transfection with miR-34a or miR-200 sponge vector (** *p* < 0.01 compared with control, *t*-test).

**Figure 3 ijms-22-09589-f003:**
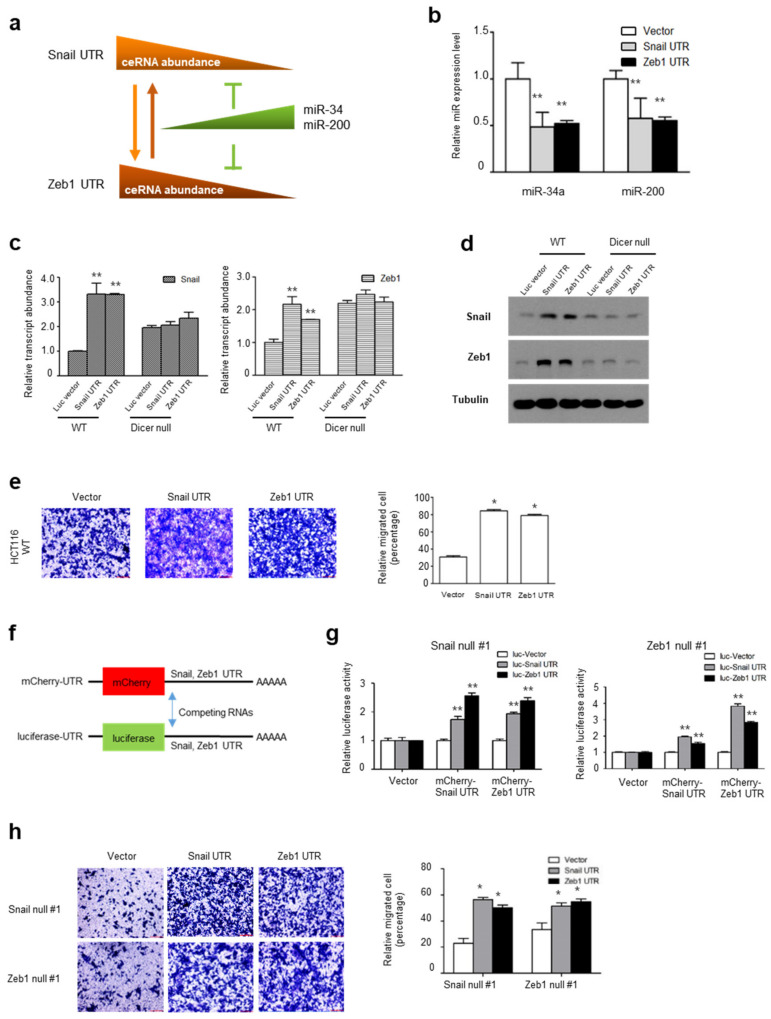
UTRs could play a role in ceRNA regulation: (**a**) Proposed schematic model of regulation between a ceRNA set and miR family; (**b**) Relative miR-34a and miR-200 expression levels in HCT116 cells following transfection with UTR expression vector of Snail and Zeb1; (**c**) qPCR analysis of Snail and Zeb1 mRNA expression in HCT116 wt and Dicer-null cells expressing Snail and Zeb1 UTR; (**d**) Immunoblot analysis of Snail and Zeb1 in HCT116 wt and Dicer-null cells expressing Snail and Zeb1 UTR (** *p* < 0.01 compared with vector control, *t*-test). Representative blots are shown from at least two independent experiments; (**e**) Representative migration images (left panel) and the number of migration cells (right panel) in HCT116 cells expressing Snail and Zeb1 UTR. The number of migration cells. Results are representative of five independent experiments (* *p* < 0.05 compared with control, Mann–Whitney test); (**f**) Schematic representation of the experimental ceRNA effects on UTRs; (**g**) Luciferase activity in Snail or Zeb1-null HCT116 cells transfected with control or UTR vector (** *p* < 0.01 compared with control, *t*-test); (**h**) Representative migration images (left panel) and the number of migration cells (right panel) in Snail or Zeb1-null HCT116 cells (* *p* < 0.05 compared with control, Mann–Whitney test).

**Figure 4 ijms-22-09589-f004:**
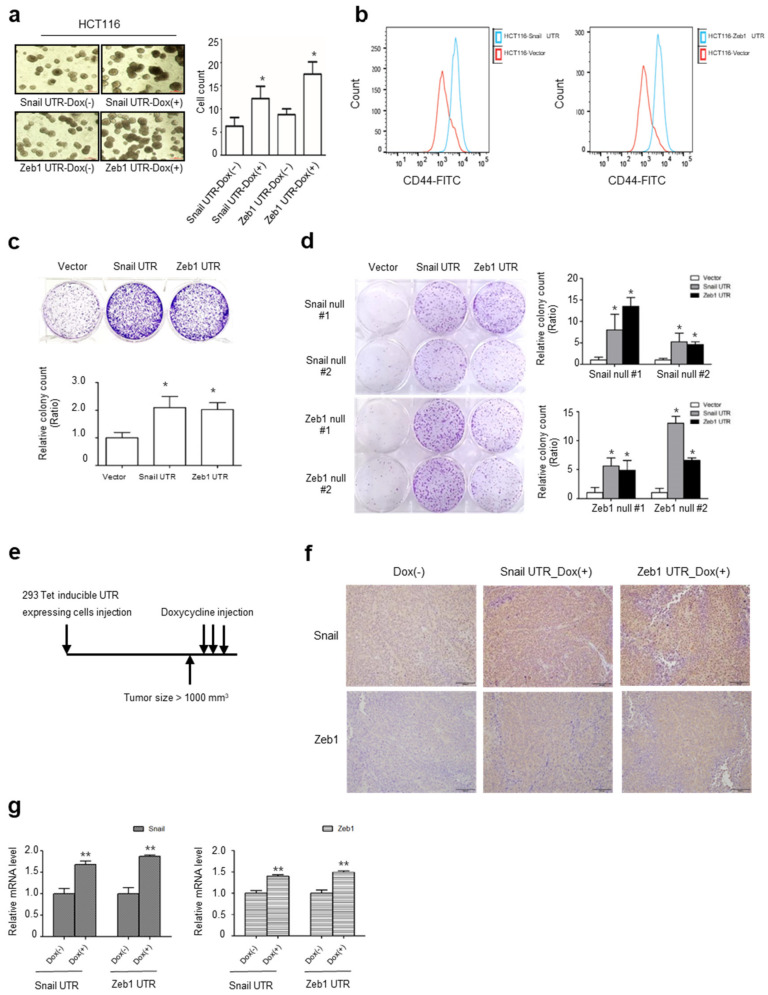
UTRs induce cancer stem cell phenotype and therapeutic resistance: (**a**) Representative images (left panel) and colony number (right panel) of UTR-expressing HCT116 cells grown in soft agarose. Cell lines were transduced with the Tet-inducible UTR expressing lentivirus, then treated with or without doxycycline (* *p* < 0.05 compared with control, Mann–Whitney test); (**b**) Flow cytometry analysis of CD44 protein in Snail UTR (left panel) or Zeb1 UTR (right panel) expressing HCT116 cells; (**c**) Clonogenic survival assay of HCT116 after transfection of Snail and Zeb1 UTR against 5 nM paclitaxel treatment (* *p* < 0.05 compared with control, Mann–Whitney test). Colonies of more than 50 cells were counted after crystal violet staining; (**d**) Clonogenic survival assay against paclitaxel treatment after transfection of UTR in Snail and Zeb1-knocked out HCT116 (* *p* < 0.05 compared with control, Mann–Whitney test); (**e**) Schematic diagram of experimental design in vivo. A doxycycline-inducible 293-UTR cell line was used. Once tumor size reached 1000 mm^3^, we induced UTR with administration of doxycycline (5 mpk) 3 times; (**f**) Snail and Zeb1 immunohistochemical staining; (**g**) Relative mRNA expression level of Snail and Zeb1 (** *p* < 0.01 compared with control, *t*-test).

**Figure 5 ijms-22-09589-f005:**
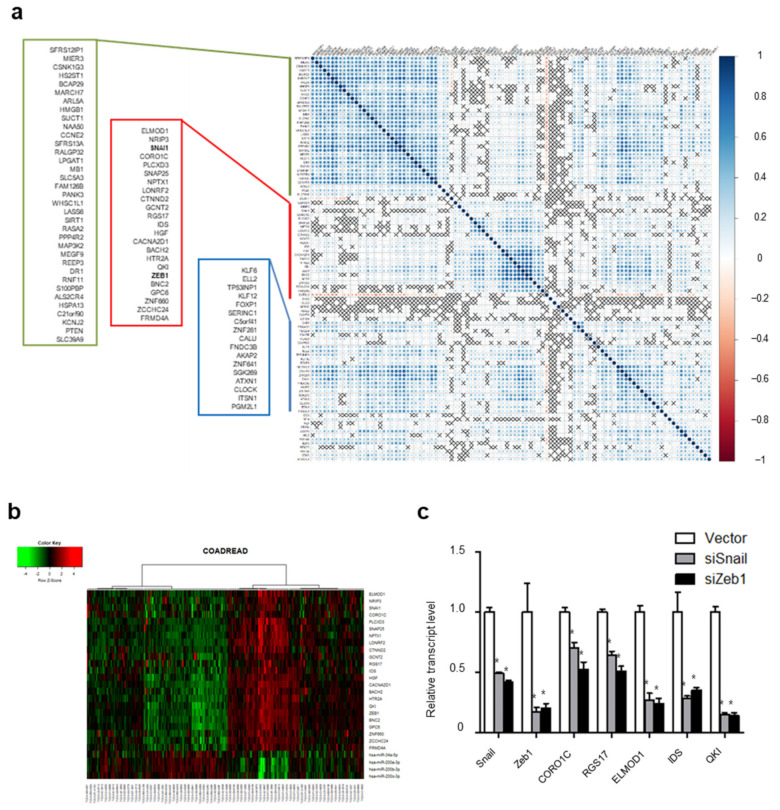
Genome-wide EMT ceRNA network: (**a**) Genome-wide correlation map associated with 100 miR-34a and miR-200 co-target genes; (**b**) Unsupervised hierarchical clustering of mRNA and tumor suppressive miRs associated with miR-34a and miR-200 co-target genes. In the heatmap, red denotes higher relative expression, whereas green indicates lower relative expression, with degree of color saturation reflecting the magnitude of the log expression signal; (**c**) qPCR of candidate genes subject to ceRNA regulation by Snail and Zeb1 in HCT116 cells (* *p* < 0.01 compared with control, *t*-test).

**Table 1 ijms-22-09589-t001:** qPCR primer sequence.

Gene	Accession Number	Primer Sequences from 5′ to 3′
Snail	NM_005985	Forward TCTCTGAGGCCAAGGATCTC
		Reverse CTTCGGATGTGCATCTTGAG
Zeb1	NM_001128128	Forward AGACATGTGACGCAGTCTGG
		Reverse TTGCAGTTTGGGCATTCATA
CORO1C	NM_014325	Forward GAAAGCACATGAAGGAGCAAG
		Reverse TGCATATTTTTCGGATTCCAG
RGS17	NM_012419	Forward AGGTCCTAGAGGAATGCCAAA
		Reverse TCTGTTCGGAGGAACTCTCTG
ELMOD1	NM_018712	Forward ATGGGAGAAGAAAAGGATGGA
		Reverse TGGGGCGATATTGTAGAAATG
IDS	NM_000202	Forward CCCATGTTCCCCTGATATTCT
		Reverse CACAAGTTCCACAAGGTCCAT
QKI	NM_006775	Forward TCCGAGGCAAAGGCTCAATGAG
		Reverse GCTCTGTTCTGAGCATCTTCCAC

## Data Availability

The data presented in this study are available in this article and the Appendix A.

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
