# Peer review of "Competing Endogenous RNA of Snail and Zeb1 UTR in Therapeutic Resistance of Colorectal Cancer"

_ijms, 2021, doi:10.3390/ijms22179589_

Round 1
Reviewer 1 Report
In the article entitled “Competing endogenous RNA of Snail and Zeb1 UTR in therapeutic resistance of colorectal cancer" the authors investigate the regulatory network between UTRs of EMT genes and tumor suppressive miRs. This is an interesting study and the hypotheses are demonstrated with an appropriate experimental approach. Appropriate methodology has been employed and the conclusions appear to be justified based on the data at hand.
In order to better appreciate the data, some minor revisions are needed:
- Please, revise the figure legends using the term “representative” with reference to the reported Western blot panels. In the figure legends the number of repeats is not indicated. Please add the information where appropriate.
- In the Discussion the authors should summarize the major conclusions of the paper and then move on to focus on the potential importance of these conclusions.
Author Response
Please, check the attached file.

Reviewer 2 Report
It is an interesting study, one cell line and xerograph model were combined with TCGA data to explore UTRs of the Snail and Zeb mainly.
Intro:
ceRNA hypothesis should be introduced again in here.
Which Wnt gene?
Results:
In vivo should be italic
Discussion:
Unsure about including the Wnt genes as the authors did not explore their involvement. Current paper is interesting however it only focuses on Snail and Zeb not other genes/mechanisms.
Discussion needs a graphical/cartoon to summarise their hypothesis. More importantly, the discussion needs better conclusion, as current status is weak and limited.
Author Response
Please, check the attached file.

Reviewer 3 Report
I think the article is very interesting and highlights the importance of ceRNA networks in tumour progression and drug resistance. Many of the experiments are clear and confirm what the authors claim, but sometimes they generalise or extrapolate these results far beyond what an experiment can demonstrate.
On the other hand, although the message of the article is clear, I find many contradictions that sometimes make it difficult to follow, for example in lines 56-58 and 107/109 it is mentioned that in the Snail UTR there is no miR-200 binding site. And yet it is continuously mentioned that this miRNA and miR-34 are targets of the Snail and Zeb1 UTR (line 41, 101/102, 237/238). So, please review and explain this better because it is contradictory as it appears.
Regarding the survival analysis (kaplan-meier), I don't understand how the low/high expression groups are obtained through a percentile of a Pvalue. Could you explain this better ? On the other hand, I recommend to use a Cox, it uses the real values of expression and it is not necessary to make these low/high groups which are often very badly distributed.
It is commented (lines ~ 180) that the overexpression of the UTR of genes related to EMT increases the expression of CD44 which is a marker of resistance to paclitaxel, but no reference appears. Right after it is stated that overexpression of UTR was sufficient to increase therapeutic resistance as shown in figure 4d. Why #1 and #2 are so different ?
Author Response
Please, check the attached file
